# Fault Feature-Extraction Method of Aviation Bearing Based on Maximum Correlation Re’nyi Entropy and Phase-Space Reconstruction Technology

**DOI:** 10.3390/e24101459

**Published:** 2022-10-13

**Authors:** Zhen Zhang, Baoguo Liu, Yanxu Liu, Huiguang Zhang

**Affiliations:** 1School of Mechanical and Electrical Engineering, Henan University of Technology, Zhengzhou 450001, China; 2Henan Key Laboratory of Superabrasive Grinding Equipment, Henan University of Technology, Zhengzhou 450001, China

**Keywords:** Re’nyi entropy, phase-space reconstruction, composite fault, rolling bearing, deconvolution

## Abstract

To address the difficulty of extracting the features of composite-fault signals under a low signal-to-noise ratio and complex noise conditions, a feature-extraction method based on phase-space reconstruction and maximum correlation Re’nyi entropy deconvolution is proposed. Using the Re’nyi entropy as the performance index, which allows for a favorable trade-off between sporadic noise stability and fault sensitivity, the noise-suppression and decomposition characteristics of singular-value decomposition are fully utilized and integrated into the feature extraction of composite-fault signals by the maximum correlation Re’nyi entropy deconvolution. Verification based on simulation, experimental data, and a bench test proves that the proposed method is superior to the existing methods regarding the extraction of composite-fault signal features.

## 1. Introduction

As a core component ensuring the safety of aircraft flight, aeroengines have a complex structure and need to operate continuously for a prolonged time under extremely severe conditions such as high temperature, high pressure, high speed, high strength, and variable load. The failure of rolling bearings, as a common component of aeroengines, seriously affects the aircraft flight safety. Hence, the early fault diagnosis of key aeroengine components is of great significance to ensuring flight safety and lowering economic and life losses [1].

Given the tight arrangement and complex structure of aeroengines, it is difficult to arrange vibration-monitoring sensors at the proximal end of the core components. As a result, the dynamic response of the core component failure will be subjected to the modulation of complex transfer paths, as well as the impact of other excitations [2,3]. Moreover, the early bearing fault may appear as a composite fault with multiple coexisting faults. Due to the coupling and interference among different faults and between faults and other excitations, the identification and separation of fault-signal features are hardly achievable, which produces huge challenges regarding the fault diagnosis of aircraft bearings [4].

To separate and extract the composite bearing fault features under complex excitations, it is necessary to deal with three aspects, namely, the selection of the sensitive feature norm, the suppression method of noise and other excitations, and the separation and decoupling method of composite noise, in order to attain a preferable performance. 

Kurtosis (Sk), as a sensitive feature capable of detecting instantaneous impacts, has received extensive attention in the fault-diagnosis field in recent years. Its successful application has been seen in the diagnosis of wind-turbine gear faults [5]; the vibration source identification for offshore wind turbines [6]; and the detection of gearbox composite faults [7], planetary gearbox faults [8], and rolling bearing faults [9], achieving a good effect. Antoni [10] elaborated on the relevant theory of Sk and formally gave its mathematical definition, i.e., the energy-normalized fourth-order spectral cumulant. Later, in 2007, a fast kurtogram (Fk) method based on short-time Fourier transform or finite impulse-response bandpass filter was proposed capable of adaptively obtaining the appropriate filter center frequency and bandwidth and effectively extracting the fault features. This method has greatly promoted the Sk application in the fault-diagnosis field. Subsequently, Lee [11] introduced a weighted kurtogram approach, whereas Wang [12] adaptively determined the filter bandwidth and center frequency by maximizing the filtered signal Sk through the incorporation of a right-expanding window. Although the above methods have all improved the Fk method to varying degrees and enhanced the fault-signal monitoring ability, all of them have encountered an unavoidable problem in practical use. That is, the primary function of Sk is to find instantaneous impacts. Thus, they are excessively sensitive to random impulses. Moreover, the inevitable sporadic strong impulse signals in actual production will greatly interfere with the ability of Sk to distinguish the real fault signals, which will even lead to the filtering failure [13]. In recent years, as an extended form of Shannon entropy, Re’nyi entropy (Re), has been widely used in the estimation of signal information content [14]. Boškosk and Juričić [15] proposed a new method to diagnose gearbox faults, using Re’nyi entropy as the characteristic index. Subsequently, it is also widely used in the fault diagnosis of bearings [16,17].

Given the structural peculiarity of the rotating machinery and bearings, the bearing signals are generated essentially by some periodic or cyclic mechanisms, so the dynamic response of bearings has cyclostationary features, and their fault signals exhibit typical repetitive transient characteristics [18]. Deconvolution is effective at eliminating the influence of complex transfer paths and enhancing the impact vibration characteristics of periodic faults. At its core, this method designs a finite impulse-response (FIR) filter with the aim of maximizing the value of the post-filtering signal sensitive characteristic index. For example, the minimum entropy deconvolution (MED) proposed by Wiggins [19], which finds the optimal filter based on the filtering result Sk, enables the iterative extraction of the impact signals from the test signals with unknown transfer paths. With advantages such as requiring less parameter adjustment and possessing fast convergence, MED has been widely applied in the field of mechanical fault-diagnosis [20,21]. However, since MED aims at Sk maximization in the iterative process, the iterations are more inclined to extract unit-impulse signals with larger values, which often fall into the local optimal solution. In 1984, Cabrelli [22] proposed the optimal minimum entropy deconvolution (OMED) and proved that it was an accurate global optimal solution. To improve the sensitivity of the MED technique to periodic impulse signals, McDonald [23] proposed the maximum correlated kurtosis deconvolution (MCKD) in 2012, with which the problem of periodic impulse deconvolution is solved. On the downside, MCKD requires the presence of priori knowledge about the fault cycle, which easily falls into the local optimal solution and is susceptible to strong noise interference. Later, McDonald put forward the multipoint optimal minimum entropy deconvolution adjusut (MOMEDA) in 2017, which eliminated the discontinuity problem between input signals and addressed some problems with OMED and MCKD, achieving preferable effects. However, all of the aforementioned deconvolution methods are considered for single-fault conditions, which are ineffective in the case of impulse signals with different periods caused by composite faults.

Phase-space reconstruction (PSR) is suitable for typical nonlinear and nonstationary mechanical impulse faults, which reflect the dynamic characteristics of the system under various fault states in a high-dimensional space. Despite receiving extensive attention from scholars in the industry [2], the use of sequential deconvolution methods of phase-space reconstruction for fault diagnosis is problematic since the sequential use of multiple methods inevitably produces signal distortion. Moreover, the use of more sequential methods indicates higher unpredictability of inter-method coupling. Singular-value decomposition (SVD), as a phase-space reconstruction algorithm with zero-phase and -time shifts, has been widely applied in extracting features of early weak-fault signals [24]. Zhao [25] discussed the similarity between SVD and the wavelet decomposition in signal processing, believing that SVD has the noise-suppression characteristics, as well as decomposition and extraction characteristics.

On the basis of the above literature analysis, because the kurtosis is too sensitive to the occasional pulse signal, it affects the accuracy of fault identification. A new method is proposed that integrates the phase-space reconstruction technique into the maximum correlation Re’nyi entropy deconvolution; in this method, the Re’nyi entropy, which can better balance the sensitivity of fault and the stability of accidental noise, is chosen as the performance index, the maximum correlation Re’nyi entropy is taken as the optimization objective, and deconvolution is chosen as the optimization method, in order to solve the problem of the low signal-to-noise ratio (SNR) and the composite-fault recognition rate, while the phase-space reconstruction technique is combined with Re’nyi entropy.

The rest of the paper is organized as follows. In Section 2, the concept and process of Re’nyi entropy, the maximum correlation Re’nyi entropy, the maximum correlation Re’nyi entropy deconvolution, and the phase-space reconstruction are introduced. In the third part, the method of phase-space reconstruction combined with maximum correlation Re’nyi entropy deconvolution is introduced. In Section 3 and Section 4, the algorithm is compared with the existing algorithms by using simulation, experimental data, and the bench test. The final conclusions are provided in Section 5.

## 2. Materials and Methods

### 2.1. Re’nyi Entropy (Re)

It is assumed that the incomplete probability set of a random event *X* is $p=\left\{p_{1},p_{2},\ldots ,p_{n}\right\}$, and its overall probability sum is $\omega \left(p\right):=\underset{i}{\sum }p_{i}\leq 1$. Then, the generalized Re parameterized by order a can be defined as
(1)$$H_{\alpha }^{R}\left(p\right)=\frac{1}{1-\alpha }log_{2}\frac{\sum_{i}p_{i}^{\alpha }}{\sum_{i}p_{i}}\;\alpha >0,\alpha \neq 1$$

The parameter α in Rényi entropy can be used to make it more or less sensitive to particular segments of the probability distributions. The exponent α helps to provide flexibility by highlighting the values closer to the edges of the probability distribution [26]. The definition of generalized Re is introduced into the actual vibration detection, and the number set $x=\left\{x_{1},x_{2},\ldots ,x_{N}\right\}$ is assumed as the discrete observation of the actual vibration process. Let there be a non-negative number set $\xi =\left\{\xi_{1},\xi_{2},\ldots ,\xi_{n}\right\}$; x corresponds to $\left\{\xi_{i}\right\}\;$in a one-to-one manner and satisfies $0\leq \xi_{i}\leq 1$ and $\overset{N}{\underset{i=1}{\sum }}\xi_{i}\leq 1$. According to the definition of generalized Re, the set $\left\{\xi_{i}\right\}$ can be regarded as the probability distribution function of random variable x(t), and $\xi_{i}$ is the probability of the instantaneous amplitude $x_{i}$ for x(t). In actual production, normal bearings often produce Gaussian vibration characteristics. In the presence of a fault, the vibrations collected from defective bearings exhibit a non-Gaussian distribution due to the fault-excited relative increase in the number of large-amplitude components. Moreover, the energy of the periodic impulse signal and the level of defect-induced excitation will increase with the defect development, ultimately resulting in the change of $H_{\alpha }^{R}$ value. Hence, Re is able to monitor the bearing-health status.

To ensure the non-negativity of $\xi_{i}$, its value can be derived from the following formula:(2)$${\left(\xi_{i}\right)}_{r}=f\left(x_{i}\right)=\frac{y_{i}}{\sum_{i}y_{i}},\;y_{i}=\{\begin{array}{l} \left|x_{i}-\mu \right|,r=av\\ {\left(x_{i}-\mu \right)}^{2},r=sv \end{array}$$
where *μ* denotes the mean value of $\left\{x_{i}\right\}$ and the subscript *r* represents the conversion method. Obviously, Formula (2) satisfies two basic conditions of generalized Re: 0$<\xi_{i}<1$ and $\underset{i}{\sum }{\left(\xi_{i}\right)}_{r}\equiv 1$. By assuming *μ* = 0 without a loss of generality, the generalized Re of order α can be obtained as
(3)$${\left(H_{\alpha }\right)}_{r}=\frac{1}{1-\alpha }log_{2}\frac{\left(1/N\right)\sum_{i=1}^{N}y_{i}^{\alpha }}{{\left(\left(1/N\right)\sum_{i=1}^{N}y_{i}\right)}^{\alpha }}+\log_{2}N,\;\alpha >0,\alpha \neq 1\;$$

We can let
(4)$$\;GM_{r}^{\alpha }=\frac{\left(1/N\right)\sum_{i=1}^{N}y_{i}^{\alpha }}{{\left(\left(1/N\right)\sum_{i=1}^{N}y_{i}\right)}^{\alpha }}=\{\begin{array}{c} \frac{\left(1/N\right)\sum_{i=1}^{N}{\left|x_{i}\right|}^{\alpha }}{{\left(\left(1/N\right)\sum_{i=1}^{N}\left|x_{i}\right|\right)}^{\alpha }},r=av,\alpha >0,\\ \frac{\left(1/N\right)\sum_{i=1}^{N}{\left(x_{i}^{2}\right)}^{\alpha }}{{\left(\left(1/N\right)\sum_{i=1}^{N}x_{i}^{2}\right)}^{\alpha }},\;\;\;\;r=sv,\alpha >0 \end{array}$$

It is clear from Formulas (3) and (4) that when measuring the amplitude distribution of $\left\{x_{i}\right\}$, $GM_{r}^{\alpha }$ is equivalent to ${\left(H_{\alpha }\right)}_{r}$ and is the kernel function of ${\left(H_{\alpha }\right)}_{r}$. The two cases of Formula (4) can be written as a unified expression as
(5)$$M_{n}=\frac{\left(1/N\right)\sum_{i=1}^{N}x_{i}^{n}}{{\left(\sqrt{\left(1/N\right)\sum_{i=1}^{N}x_{i}^{2}}\right)}^{n}}=\frac{\left(1/N\right)\sum_{i=1}^{N}x_{i}^{n}}{\sigma^{n}}$$

As is clear from Formula (5),$\;GM_{r}^{\alpha }$ can be regarded as the generalization of Re. Obviously, by choosing different *r* and α values, various statistical parameters can be derived from $GM_{r}^{\alpha }$. When r=av,  a=3, r=sv*,*
a=3/2*,* and a=2, the following three statistical indicators can be derived separately:(6)$$GM_{av}^{3}=\frac{\left(1/N\right)\sum_{i=1}^{N}{\left|x_{i}\right|}^{3}}{{\left(\left(1/N\right)\sum_{i=1}^{N}\left|x_{i}\right|\right)}^{3}}=Re,\;\;\;\;\;\;\;\;\;\;\;\;\;\;\;\;\;\;\;\;\;\;\;\;\;\;\;\;\;\;\;\;\;\;\;\;\;\;\;\;\;\;\;\;r=av,\;a=3\;\;GM_{sv}^{3/2}=\frac{\left(1/N\right)\sum_{i=1}^{N}{\left(\left(x_{i}^{2}\right)\right)}^{3/2}}{{\left(\left(1/N\right)\sum_{i=1}^{N}\left(x_{i}^{2}\right)\right)}^{3/2}}=\frac{\left(1/N\right)\sum_{i=1}^{N}{\left|x_{i}\right|}^{3}}{\sigma^{3}}=Sr\;\;\;\;\;\;\;\;r=sv,\;a=3/2\;GM_{sv}^{2}=\frac{\left(1/N\right)\sum_{i=1}^{N}\left(\left(x_{i}^{2}\right)\right)}{{\left(\left(1/N\right)\sum_{i=1}^{N}\left(x_{i}^{2}\right)\right)}^{2}}=\frac{\left(1/N\right)\sum_{i=1}^{N}x_{i}^{4}}{\sigma^{4}}=Sk\;\;\;\;\;\;\;\;r=sv,\;a=2\;$$

According to Formula (6), $GM_{sv}^{2}$ is equivalent to *Sk* and $GM_{sv}^{3/2}$ is equivalent to the third-order moment Skewness (*Sr**).* Thus, $GM_{av}^{3}$ has become a new statistical indicator Re, i.e., the narrow-sense *R**e* proposed herein. This means that Re has a similar mathematical expression to Sk and *Sr*. Therefore, Re*,*
Sr*,* and Sk can be considered as different expressions derived from the generalized Re, all of which can be explained by generalized Re theory.

Suppose the fault at the inner ring of rolling bearing is
$$x\left(t\right)=\overset{M}{\underset{j}{\sum }}a_{j}e^{-g\gamma \left(t\right)}cos[\omega_{0}\sqrt{1-g^{2}}\left(\gamma \left(t\right)-\tau_{j}\right)$$where $\;a_{j}\;$ is the magnitude of the Jth fault shock, g  denotes the attenuation coefficient of the bearing, M is the number of excitation of the bearing shock,  γ(t)  is the pseudo-cycle time, T denotes the period of the bearing fault shock, $\tau_{j}\;$ stands for the time delay due to relative slip, and $\;f_{e}\;$ stands for the fault characteristic frequency. The values of the simulation parameters are shown in Table 1.

Figure 1 depicts the variations of Sk*,*
Sr*,* and Re with the fault defect at the inner ring of the rolling bearing. To facilitate understanding and display, the defect evolution is converted into the SNR change. According to Figure 1, Sr is insensitive to the defect size, while Re and Sk share similar variation trends. Thus, clearly, Re and Sk are highly sensitive to the alterations of fault defects, which can rather accurately indicate the bearing faults. Since this phenomenon may present similar variation trends for the outer ring and the rolling element faults, it is not described in detail here.

Figure 2 depicts the variations of three performance indicators —Sk*,*
Sr*,* and Re—with the sporadic impulse-response. Data on the figure demonstrates that Sr has excellent robustness to the sporadic impulse-response, whereas Sk is highly sensitive to the sporadic impulse-response. The Sk value increases by over three-fold upon sporadic interference, indicating that the aero-engine produces a greater impact on Sk when it has sporadic impulses. Re is somewhat sensitive to sporadic impulses, although the overall variation is not large. As suggested by Figure 1 and Figure 2, Re has a preferable ability to trade off between the bearing defects and the sporadic impulse sensitivity.

### 2.2. Correlation Re’nyi Entropy (CRe)

Statistical properties of rotating components such as bearings, gears, shafts, and propellers change periodically over time. Thus, the signals generated by such components are called cyclostationary signals. Conventional signal processing can extend and exploit this feature. A signal can be nth-order cyclostationary when its nth-order statistic is periodic [27]. To exploit the cyclostationarity of rotating components, this study defines the correlation Re’nyi entropy (CRe) deconvolution on the basis of Re, with a view to extracting periodic impulse signals. The first-order CRe and *M*th-order CRe are defined separately.
(7)$$CRe^{1}\left(T\right)=\frac{\sum_{n=1}^{N}{\left(y_{n}y_{n-T}\right)}^{\frac{3}{2}}}{{\left(\sum_{n=1}^{N}y_{n}\right)}^{3}}$$
(8)$$CRe^{M}\left(T\right)=\frac{\sum_{n=l}^{N}{\left(\prod_{m=0}^{M}y_{n-mT}\right)}^{\frac{3}{2}}}{{\left(\sum_{n=1}^{N}y_{n}\right)}^{M+2}}$$

Noise signal, sporadic impulse + noise signal, cosine signal + noise signal, and periodic impulse + noise signal (Figure 3) are defined to analyze Sk, $Re,\;CSk^{1}$*,* and $CRe^{1}$. Since Gaussian noise signals are stable for the same higher-order statistics, the normalized sensitivity $\rho_{ij}$ is defined to explain the sensitivity of the aforementioned indicators to typical signals.
(9)ρij=pijp0ji=[0,1,2,3]=[a,b,c,d];      j=[0,1,2,3]=[Sk,Re,CSk1,CRe1]

It is clear from Figure 4 that Sk has high sensitivity to both sporadic and periodic impulse signals and is easily affected by sporadic noise. CSk [20] and CRe have good robustness to both harmonic and sporadic impulse signals, and they are highly sensitive to periodic impulse signals, which can thus distinguish the periodic impulse signals. A horizontal comparison reveals that CRe has better sensitivity to CSk under the periodic impulse condition.

### 2.3. Maximum Correlation Re’nyi Entropy Deconvolution (MCReD)

Given the complex and compact interior of aeroengines, the vibration sensors cannot be arranged near the faulty bearings since the fault signals are easily affected by strong noise and intricate transmission paths. Thus, the dynamic response of faulty bearings can be regarded as the linear convolution of vibration signal and channel. The deconvolution method can effectively eliminate the influence of intricate transmission paths and enhance the fault-impact vibration characteristics. In this study, a novel MCReD-based method is proposed to overcome the limitations of heavily Gaussian and non-Gaussian background noises.

An actual fault signal of the aeroengine bearing will contain multiple components, which can be expressed as
(10)$$x\left(t\right)=h_{d}*d+h_{u}*u+h_{e}*e\;x=\left[\begin{array}{c} x_{1}\\ x_{2}\\ \vdots \\ x_{N} \end{array}\right],\;d=\left[\begin{array}{c} d_{1}\\ d_{2}\\ \vdots \\ d_{N} \end{array}\right],\;u=\left[\begin{array}{c} u_{1}\\ u_{2}\\ \vdots \\ u_{N} \end{array}\right],\;e=\left[\begin{array}{c} e_{1}\\ e_{2}\\ \vdots \\ e_{N} \end{array}\right]$$
where x stands for the vibration signal acquired by remote vibration sensor; d stands for the impulse signal generated by a faulty aero-engine bearing; u represents the interference signal generated by the other aero-engine components; e refers to the background noise; and $h_{d},\;\;h_{u},$ and $h_{e},$ respectively, stand for the transfer functions corresponding to different inputs.

The core idea of the MCReD algorithm is to find the global optimal FIR filter $\vec{f}$ by the deconvolution operation on the basis of eliminating the background noise and other interfering elements to the maximum extent, in order to highlight the pulse sequence in the fault signal, it can be expressed mathematically as:
(11)$$\left\{ \begin{array}{l} \vec{y}=\vec{f}*\left(\vec{h_{u}}*\vec{u}\right)+\vec{f}*\left(\vec{h_{d}}*\vec{d}\right)+\vec{f}*\left(\vec{h_{e}}*\vec{e}\right)\\ \text{St.}\;\vec{f}*(\vec{h_{u}}*\vec{u})\to 0\;\;\;\;\;\;\;\;\;\;\;\;\;\;\;\;\;\;\;\;\;\;\;\;\;\;\;\;\;\;\;\;\;\;\;\;\;\;\;\;\;\;\;\;\;\;\;\;\;\;\;\;\;\;\;\;\;\;\;\;\;\;\;\;\;\;\;\;\mathrm{where}\;\vec{f}={\left[f_{1}\;f_{2}\;\ldots f_{L}\right]}^{T}\\ \vec{f}*\left(\vec{h_{d}}*\vec{d}\right)+\vec{f}*(\vec{h_{e}}*\vec{e})\to 0 \end{array}\right.$$

From Formula (7) and the discrete signal convolution formula, the following can be deduced:(12)$$y_{n}=\sum_{k=1}^{L}f_{k}x_{m},\;m=n-k+1$$

Without a loss of generality, the relevant conclusion can be illustrated by the Formula (7) for first-order *Cre* (M = 1). The optimal filter can be obtained by the following formula:(13)Cf   →maxRe1(T)=   f   →max∑n=1N(ynyn−T)32(∑n=1Nyn)3, f→=[f1 f2 …fL]T

To solve the filter coefficient of maximum *CRe*, Formula (13) can be solved and expressed as
(14)$$\frac{d}{df_{k}}CRe^{1}\left(T\right)=0,\;k=1,2,3,\ldots L$$

Calculating the derivative $CRe^{1}N$ of Formula (14)’s denominator and substituting the $\frac{d}{df_{k}}y_{n}=x_{m}$*,*
m=n−k+1 yield:(15)$$\begin{array}{c} \frac{d}{df_{k}}CRe^{1}N=\frac{3}{2}\overset{1}{\underset{m=0}{\sum }}X_{mT}\vec{a}\;\\ \;\mathrm{where}\;X_{i}={\left[x_{1+i-r},\;x_{2+i-r},\ldots ,x_{N+i-r}\right]}^{T},\;\;\;X_{r}=\left[X_{1},X_{2},\ldots ,X_{L}\right]\\ \;\;\;\;\;\;\;\;\;\;\;\;\;\;\;\;\;\;\;\;\;\;\;\;\;\;\;\;\;\;\;\;\;\;\;\;\;\;\;\vec{a}=\left[\begin{array}{c} y_{1-mT}^{-1}{\left(y_{1}y_{1-T}\ldots \;y_{1-MT}\right)}^{\frac{3}{2}}\\ y_{2-mT}^{-1}{\left(y_{2}y_{2-T}\ldots \;y_{2-MT}\right)}^{\frac{3}{2}}\\ \vdots \\ y_{N-mT}^{-1}{\left(y_{n}y_{n-T}\ldots \;y_{n-MT}\right)}^{\frac{3}{2}} \end{array}\right] \end{array}$$

Similarly, calculating the derivative $\frac{d}{df_{k}}CS_{1}D$ of numerator yields
(16)$$\frac{d}{df_{k}}CRe^{1}D=3{\left(\sum_{n=1}^{N}y_{n}\right)}^{2}x_{m}$$

By synthesizing Formulas (14) and (15), the following can be obtained according to Formula (13)
(17)$$\begin{array}{l} \frac{d}{df_{k}}CRe^{1}\left(T\right)=3{\left(\sum_{n=1}^{N}y_{n}\right)}^{2}CRe^{1}N-6{\left\|\vec{b}\right\|}^{3}\sum_{n=1}^{N}y_{n}\left(x_{m}\right)\\ \;\;\;\;\;\;\;\;\;\;\;\;\;\;\;\;\;\;\;\;\;\;\;\;\;\;\;\;\;\;\;\;\;\;\;\;\;\;\;\;\;\;\;\;\;\;\;\;\;\;\;\;\;\;\;\;\mathrm{where}:\vec{b}={\left[\begin{array}{c} y_{1}y_{1-T}\;\ldots \;y_{1-MT}\\ y_{2}y_{2-T}\;\ldots \;y_{1-MT}\;\;\\ \vdots \;\;\\ {\begin{array}{c} y \end{array}}_{N}y_{N-T}\ldots \;y_{N-MT}\; \end{array}\right]}^{T} \end{array}$$

Additionally, since
$$\vec{y}=X_{0}^{T}\vec{f}$$
the arrangement yields
(18)$${\left(\sum_{n=1}^{N}y_{n}\right)}^{2}\sum_{m=0}^{1}X_{mT}\vec{a}=2{\left\|b\right\|}^{3}X_{0}\vec{y}$$

If ${\left(X_{T}X_{0}^{T}\right)}^{-1}$ exists, the solution formula for first-order $CRe^{1}$ can be obtained as
(19)$$\vec{f}=\frac{{\left(\sum_{n=1}^{N}y_{n}\right)}^{2}}{2{\left\|\vec{b}\right\|}^{3}}{\left(X_{0}X_{0}^{T}\right)}^{-1}\sum_{m=0}^{1}X_{mT}\vec{a}$$

This formula can be generalized to the Mth-order as:
(20)$$\vec{f}=\frac{{\left(\sum_{n=1}^{N}y_{n}\right)}^{2}}{2{\left\|\vec{b}\right\|}^{3}}{\left(X_{0}X_{0}^{T}\right)}^{-1}\sum_{m=0}^{M}X_{mT}\vec{a}$$

### 2.4. Phase-Space Reconstruction(PSR)

MCReD can enhance the periodic impulse signals and eliminate the negative effects of transfer function and sporadic impulse-response. However, similar to the case of the MCKD method, its ability to extract composite faults with different periods is unsatisfactory at low SNR.

PSR, as a time-series analysis technique, recovers important system-component information from the high-dimensional space that is extended from a one-dimensional time series. The phase-space trajectory matrix $X_{r}$ composed of original signals is a Hankel matrix, which can be reconstructed by selecting the components of fault information based on the noise-suppression and decomposition characteristics of SVD.

For a phase-space trajectory matrix $X_{r}$*∈*$R^{L\times N}$, regardless of whether its rows and columns are correlated or not, there must be orthogonal matrices ***U*** = $u_{1},u_{2}\ldots u_{n}\in R^{L\times N}$ and $V\;=\;v_{1},v_{2}\ldots v_{n}\in R^{L\times N}$*,* so that
(21)$$X_{r}=USV_{T}=\overset{i=L}{\underset{i=1}{\sum }}\delta_{i}u_{i}v_{i}=\overset{i=L}{\underset{i=1}{\sum }}A_{i}$$
is established. In the formula, the diagonal matrix is ***S*** = diag[$\delta_{1}$, $\delta_{2}$*…*$\delta_{m}$], where $\delta_{1}$ *≥*
$\delta_{2}$
*≥ … ≥*
$\;\delta_{m}$
*>* 0. Formula (20) is called the SVD of *x(t)*. The main diagonal element of **S** is the singular value of the matrix $X_{r}$. $A_{i}$ is defined as the subspace of matrix **X**, and the signal $x_{\begin{array}{c} i \end{array}}^{\prime }\left(l\right)$ reconstructed by the anti-diagonal method is defined as the sub-signal discretization.

In PSR, different subspaces can be selected for reconstruction to represent different signal characteristics. After the selection of sub-spaces, signals can be reconstructed to obtain a reconstructed phase-space that can represent the signal characteristics.
(22)$$X_{r}^{c}=\sum_{i\in P}\sigma_{i}u_{i}v_{i}$$
where P stands for the sub-signal screening 

Substituting the reconstruction matrix into Formula (19) yields
(23)$$\vec{f}=\frac{{\left(\sum_{n=1}^{N}y_{n}\right)}^{2}}{2{\left\|\vec{b}\right\|}^{3}}{\left[X_{0}^{c}{\left(X_{0}^{c}\right)}^{T}\right]}^{-1}\sum_{m=0}^{M}X_{mT}^{c}\vec{a}$$

After the integration of the PSR, substantial priori knowledge required by MCReD will no longer be important. The rather important filter length L and period T in MCKD, for instance, only require suitable value-taking in PSR-MCReD due to the noise-suppression, decomposition, and extraction characteristics of PSR, where no accurate fault eigenperiod is needed. Thus, PSR-MCReD also has a certain blind-solution property.

### 2.5. Determination of Screening Space

With the PSR algorithm, screening of the subspace containing more fault information for reconstruction is critical. Using the sub-signal Re as the performance indicator, this study carries out reconstruction by selecting the subspace that contains subsignal mapping with a maximum Re value.

Phase-space trajectory matrix *X_r_* is subjected to SVD.*A_i_* values of various orders are derived separately, and the anti-diagonal method is used to reconstruct the sub-signals $x_{\begin{array}{c} i \end{array}}^{\prime }\left(l\right)$ of various orders for *A_i_*.$Re_{i}=Re\left(x_{\begin{array}{c} i \end{array}}^{\prime }\left(l\right)\right)$ is calculated and arranged in descending order as [*Re_i_*_1_, *Re_i_*_2_, …, *Re_ij_*], *Re_i_*_1_ ≥ *Re_i_*_2_ ≥ … ≥ *Re_ij_* > 0.By adopting the difference method *D_j_*_−1_ = *Re_ij_* − *Re_ij−_*_1_ and solving max(*D_j_*_−1_), the number of elements in the *P* space is determined to be *j* − *1*.Screening space P is obtained based on the *i* mapped by *Se_ij_*.

## 3. PSR–MCReD-Based Fault-Diagnosis Method

In the phase-space reconstruction, the key of the phase-space reconstruction algorithm is to filter the subspace that contains more fault information. In this paper, the Re’nyi entropy of the subsignal is taken as the performance index, and the subspace containing the maximum Re’nyi entropy subsignal mapping is selected for reconstruction. The specific flowchart is shown in Figure 5.

Fault signal is input.$X_{r}^{c}$ is calculated according to the screening space selection method for PSR.Filter length is assumed as L, and to prevent the filter from falling into local optimal solution, the initial filter is assumed to be $\vec{f}={\left[0\;0\ldots \;1\;1\;\ldots \;0\;0\right]}^{T}$.$\vec{y}=X_{0a}^{T}\vec{f}$, the filtered signal $\vec{y}$ is calculated.$\vec{a}$ and $\vec{b}$ are calculated according to Formula (14).Filter $\vec{f}$ is updated according to Formula (19).If $\nabla \frac{d}{df_{k}}CS_{M}\left(T\right)>\varepsilon$ is determined to be true, step 4 is returned to; if false, the process is ended. To avoid the iteration falling into an infinite loop, the ε in the formula is chosen as a tiny positive number.Final filter order result is calculated by formula $\;\vec{y}=X_{0a}^{T}\vec{f}$.Envelope spectrum analysis is performed, and fault information is obtained.

## 4. Results

### 4.1. Simulation Analysis

Suppose that the faulty bearing has both inner and outer ring faults and is disturbed by signals such as random impacts, discrete harmonics, and Gaussian white noise.
$$\left\{ \begin{array}{l} x\left(t\right)=x_{1}\left(t\right)+\mu x_{2}\left(t\right)+x_{3}\left(t\right)+x_{4}\left(t\right)+n(t)\\ x_{1}\left(t\right)=\sum_{-\infty }^{+\infty }A\left(KT_{1}\right)e^{-\varepsilon \left(t-KT_{1}\right))}sin[2\pi f_{s}(t-KT_{1}+\varphi_{k1})]\\ x_{2}\left(t\right)=\sum_{-\infty }^{+\infty }e^{-\varepsilon \left(t-KT_{2}-\frac{T_{2}}{2}\right))}sin\left[2\pi f_{s}\left(t-KT_{2}--\frac{T_{2}}{2}+\varphi_{k2}\right)\right]\\ x_{3}\left(t\right)=\sum_{i=1}^{M_{1}}D_{i}e^{-\varepsilon \left(t-iT_{r}\right))}sin\left[2\pi f_{s}\left(t-iT_{r}+\varphi_{k3}\right)\right]\\ x_{4}\left(t\right)=\sum_{j=1}^{M_{2}}p_{j}sin\left(2\pi f_{4j}t-\varphi_{k4}\right) \end{array}\right.$$


$x_{1}\left(t\right)$ denotes the dynamic response of bearing with the inner ring fault, $x_{2}\left(t\right)$ denotes the dynamic response of bearing with the outer ring fault, $x_{3}\left(t\right)$ represents the external random impact interference during the measurement, $x_{4}\left(t\right)$ stands for the discrete harmonics from shafts or other components received by the remote sensors, and n(t) represents the Gaussian white noise with SNR= −8. Table 2 details the specific parameter selection.

Figure 6 displays the time domain waveforms of various component signals.

To evaluate the performance, this section compares the proposed FSR-MCReD method with the MCKD and FK methods [26] using simulation-signals, where the MCKD and FSR-MCReD filter lengths are *L*= 100, and the number of iterations is set at 30.

It is clear from Figure 7 that under complex operating conditions, the Sk value of random impulses is often larger than the periodic impulse sequence of signals, so that the filter fails to perform filtering in the frequency bands selected by the FK method, leading to the failure of fault identification. According to Figure 8, the filtering effect with the MCKD method is greatly compromised under the composite-fault condition, where the effective fault identification is hardly achievable. Figure 9a displays the differential Re result of PSR signals by the FSR-MCReD method when M = 1. As is clear, the differential Re values for two cycles are similar, with the maximum values both being 2. According to Figure 9b, since FSR-MCReD adopts the Re as the optimization condition, which is more robust to periodic simple harmonics and sporadic impulses, the decomposition and noise-suppression characteristics of PSR are fully exploited and integrated into the MCReD calculation, thereby enabling the preferable identification of composite faults.

### 4.2. Aeroengine Fault-Diagnosis

Bearings are the core components of aeroengines. Since the aeroengines are often tightly arranged, the vibration sensors can hardly be installed at the proximal end, which causes great difficulties in the feature-extraction of fault signals. Aeroengine vibration data were acquired from an accessory gearbox (Safran, France) [28]. Figure 10 displays the structural schematic of the aeroengine and the locations of sensors. At the L5 shaft location, outer ring spalling damage and retainer failure were present for the rolling bearing structure. Vibration sensors were arranged near the L1 and L5 shafts of the gearbox, while the rotational speed sensors were arranged on the L4 shaft. Measured data of vibration sensor 2 was selected for analysis, and the rotational speed and fault eigenfrequency were calculated according to the L4 shaft speed, the numbers of teeth on L4 and L5 shafts, and the bearing parameters of L5 rolling bearing (Table 3). Through comparison with the FK and FSC [29] algorithms, the ability of FSR-MCReD to extract features of aeroengine fault signals was verified.

Figure 11 displays the envelope spectrum analysis results with the FK algorithm. As is clear, the FK algorithm has a good filtering effect. However, filtering errors are caused due to excessive and complex external noise, making it difficult to distinguish the fault features of the outer ring signals. Figure 12 presents the analysis results with the FSC algorithm. Clearly, the FSC algorithm has a good enhancement effect on the periodic signals. However, since it has no filtering effect, the fault features of outer ring signals are covered in substantial irrelevant signal features, making it difficult to diagnose faults accurately. Figure 13 displays the envelope analysis results with the FSR-MCReD. As is clear, after integration of PSR into the MCReD algorithm, both the signal filtering and decomposition effects are improved, thus enabling accurate fault-diagnosis.

### 4.3. Validation of Composite-Fault Experimental Data

The accelerated life cycle test data of bearings provided by Professor Lei’s team from Xi’an Jiaotong University [30] enriches the experimental data on the fault-diagnosis and performance-degradation research. Under the experimental conditions of composite outer and inner ring faults, this study used the experimental data collected at t = 20 min, which were not clearly characterized by the CMS index, to theoretically calculate the fault frequency $f_{i}=171\;\mathrm{Hz},\;f_{o}=109\;\mathrm{Hz}$. Further, a comparative analysis was made against the deconvolution algorithms OMEDA and MOMEDA [2], in order to demonstrate the advantages of our proposed algorithm in solving composite faults.

Figure 14 presents the envelope spectrum analysis results with the OMED, the MOMEDA, and our FSR-MCRD methods. As demonstrated by the results, both OMED and MOMEDA can preferably analyze the outer ring faults having obvious failures, with the MOMEDA exhibiting a better filtering effect since it addresses the local optimal solution problem in the OMED method. The common defect of the two, however, is that the inner ring faults are not displayed preferably. The method proposed herein, on the other hand, allows for better extraction of the fault signals in composite faults due to the incorporation of FSR’s noise-suppression and decomposition characteristics into the MCRD method. Thus, it has better diagnostic ability for composite faults.

### 4.4. Experimental Verification

Regarding the experimental setup, the rotating machinery fault test bench from Spectra Quest, the acceleration sensor from Yangzhou Kedong, the displacement sensor from Bently, and the LMS data acquisition system were adopted. The rotational speed was set at 2700 rpm, and the faulty bearing model was MB ER-12 K. Composite-fault bearings with a ball fault, an inner ring fault, and an outer ring fault were set up separately. The rotational speed was kept constant at 2700 rpm, and the vibration signals of acceleration sensors were collected by the LMS SCADAS mobile data acquisition system at a sampling frequency of 25.6 kHz. Table 4 details the eigenfrequency calculations of various faults, whereas Figure 15 depicts the experimental panorama. MCKD, OMED, MOMOEDA, and FSR-MCRD were separately used for comparison.

Figure 16 displays the envelope spectrum analysis results with the aforementioned four methods. As is clear, the MCKD method has a good effect on signal feature-extraction under single-fault conditions, which, however, easily falls into a local optimum. Thus, it hardly achieves feature extraction or enhancement for other coexisting periodic impulse signals. The non-iterative method OMED exhibits the worst effect of fault feature-extraction, which can hardly diagnose faults. In the case of MOEDA, the results of fault feature-extraction are incomplete since it is an improved version of OMED, which does not consider the condition of multiple fault coexistence. With the FSR-MCReD method, the simultaneous existence of multiple faults is considered, and Re, which has good stability for both sporadic impulses and harmonics, is used as the performance index. Moreover, PSR is integrated for performing deconvolution, so that the subspace signals containing more fault information are extracted, thus preferably achieving the feature extraction under composite-fault condition. However, it is noteworthy that the second- and third-order inner ring fault features are lost during feature extraction since part of the subspace is truncated in PSR.

## 5. Conclusions

This paper indicatively introduces the Re’nyi entropy, which can better balance the fault sensitivity and the stability of the accidental pulse signal, into the fault diagnosis of an aero-engine. The signal transmission noise and accidental noise are solved by MCReD. In order to better decouple the complex fault signals and improve the extraction ability of MCReD under low SNR conditions, the PSR technique is integrated into the MCReD, not a simple combination. The proposed method has achieved good results in simulation and experimental verification.

In this study, a novel method for extracting the fault-signal features of aeroengine core components based on maximum CRe deconvolution integrating PSR is proposed. Through simulation analysis, experimental data, and a bench test, the advantages of the proposed method over the existing signal feature-extraction methods are verified regarding the extraction of complex fault-signal features under low SNR and complex noise interference.The mathematical formulas for Sk, Sr, and the narrow-sense Re proposed in this study are deduced based on the generalized Re. The three performance indicators can all be regarded as different expressions for generalized Re. As revealed by a simulation experiment on the correlations of the three with fault sensitivity and sporadic noise stability, both Re and Sk are highly sensitive to the faults, and Re is more stable to sporadic noise.Inspired by CK, a definition of CRe is given, which has a better suppression effect on the sporadic and harmonic noises. By deriving the solution formula of MCReD, a non-iterative method for solving the MCReD of the optimal filter is proposed.For the impulse signals with different periods under composite-fault conditions, a maximum Re-based subspace-filtering method is proposed by integrating the PSR technique into the MCReD calculation, which utilizes the noise-suppression and decomposition characteristics of SVD.Through simulation, experimental data validation, and bench test verification, the method proposed herein is proven to be fairly effective at extracting composite-fault signal features under low SNR and complex noise conditions.

In this paper, firstly, the maximum correlation Re’nyi entropy deconvolution algorithm is similar to the maximum correlation kurtosis deconvolution algorithm, which is dependent on prior knowledge. Secondly, the theoretical derivation of the decoupling characteristics of the phase-space technology needs to be improved, and the selection method of the screening space of the singular-value decomposition needs to be further optimized. Finally, some experiments on the application of PSR-MCReD are carried out, and more experiments need to be completed in the future work.

## Figures and Tables

**Figure 1 entropy-24-01459-f001:**
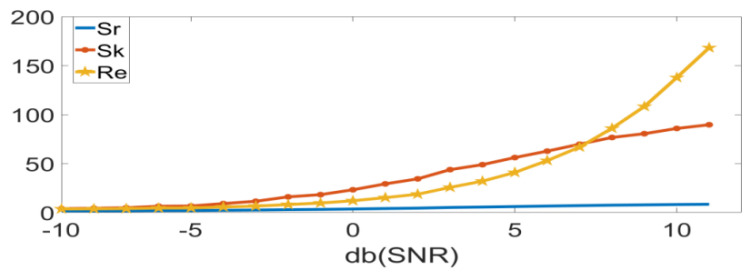
Sk, Sr, Re schematic diagram of the variation with defects.

**Figure 2 entropy-24-01459-f002:**
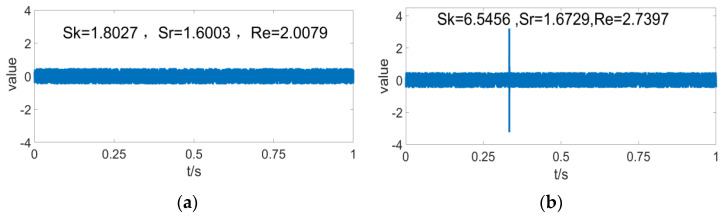
Sk, Sr, *R*e sensitivity to accidental noise; (**a**)  Sk, Sr, Re under noise condition; (**b**) Sk, Sr,  *R*e under noise and occasional pulse conditions.

**Figure 3 entropy-24-01459-f003:**
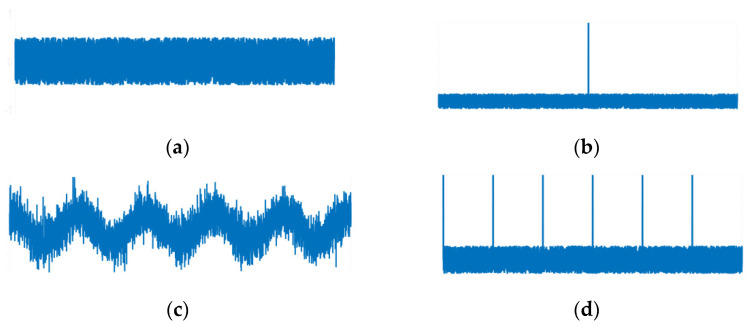
Typical signal of (**a**) noise condition, (**b**) occasional pulse with noise, (**c**) harmonic signal with noise, and (**d**) periodic pulse with noise.

**Figure 4 entropy-24-01459-f004:**
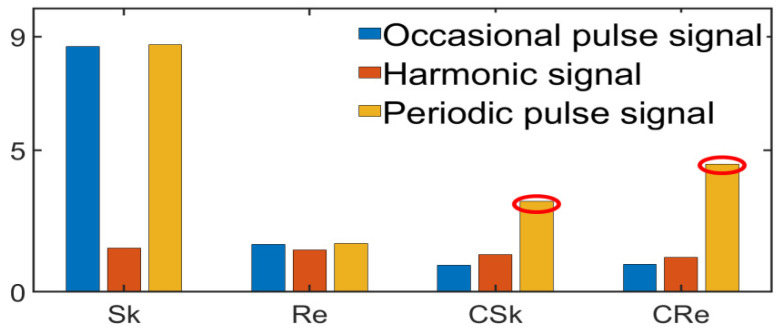
Normalized sensitivity.

**Figure 5 entropy-24-01459-f005:**
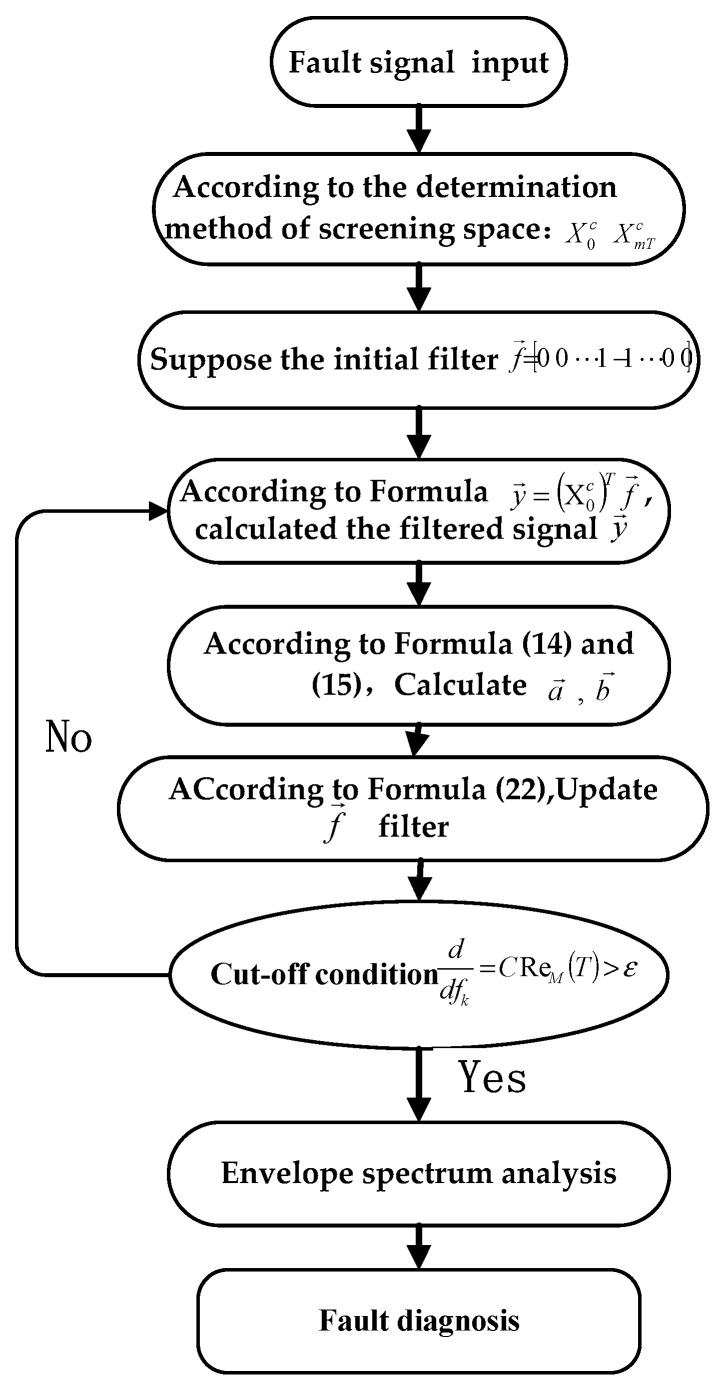
Flowchart of PSR-MCReD.

**Figure 6 entropy-24-01459-f006:**
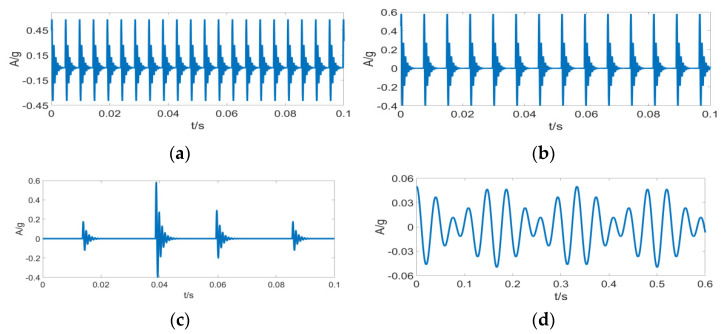
Component signal time domain waveform of (**a**) inner fault $x_{1}\left(t\right)$ time domain waveform, (**b**) outer fault $x_{2}\left(t\right)$ time domain waveform, (**c**) outer fault $x_{2}\left(t\right)$ time domain waveform, and (**d**) random pulse $\;x_{3}\left(t\right)$ time domain waveform.

**Figure 7 entropy-24-01459-f007:**
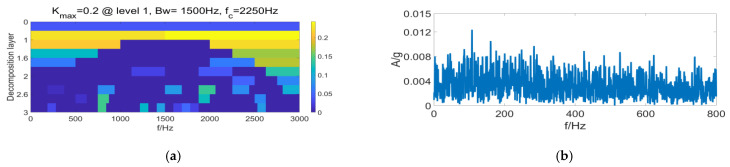
Fast spectral kurtosis results of (**a**) fast kurtosis, and (**b**) fast spectral kurtosis envelope spectrum.

**Figure 8 entropy-24-01459-f008:**
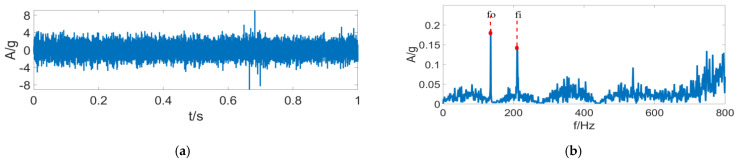
MCKD results of (**a**) time-domain results of MCKD, and (**b**) FSR-MCReD envelope spectrum.

**Figure 9 entropy-24-01459-f009:**
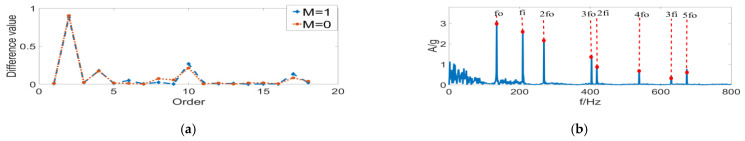
FSR-MCReD results of (**a**) determination of filter space, and (**b**) FSR-MCReD envelope spectrum.

**Figure 10 entropy-24-01459-f010:**
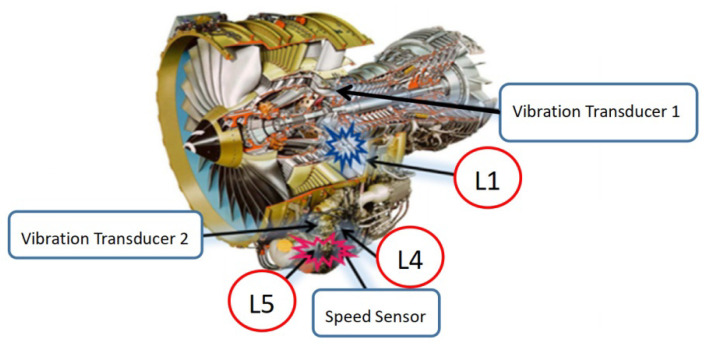
A schematic diagram of an aero-engine structure.

**Figure 11 entropy-24-01459-f011:**
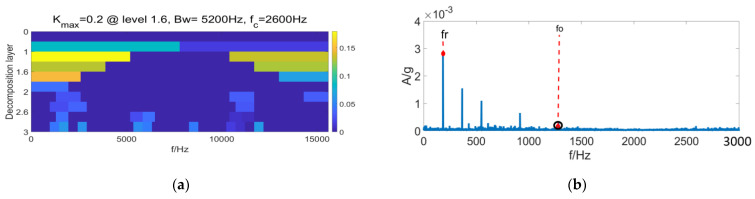
Fast spectral kurtosis results of (**a**) fast kurtosis, and (**b**) fast spectral kurtosis envelope spectrum.

**Figure 12 entropy-24-01459-f012:**
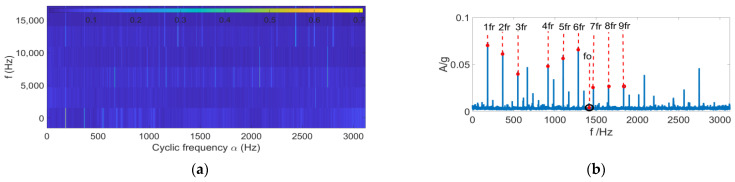
FSC results of (**a**) fast spectral correlation, and (**b**) FSC enhanced envelope spectra.

**Figure 13 entropy-24-01459-f013:**
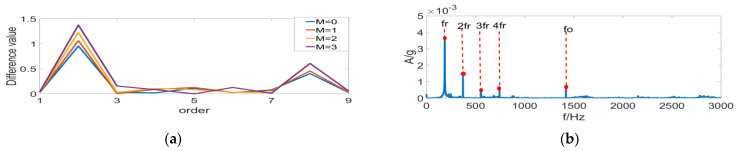
FSR-MCReD results of (**a**) determination of filter space, and (**b**) FSR-MCReD envelope.

**Figure 14 entropy-24-01459-f014:**
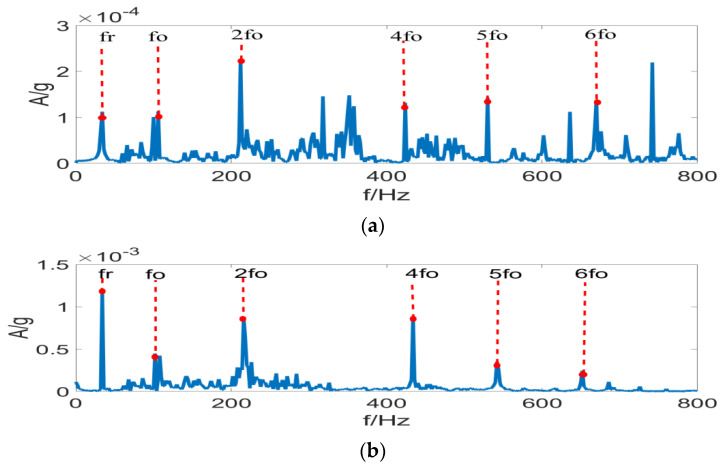
Three methods result of (**a**) OMED envelope spectra (**b**) MOMED envelope spectra, and (**c**) MOMED envelope spectra.

**Figure 15 entropy-24-01459-f015:**
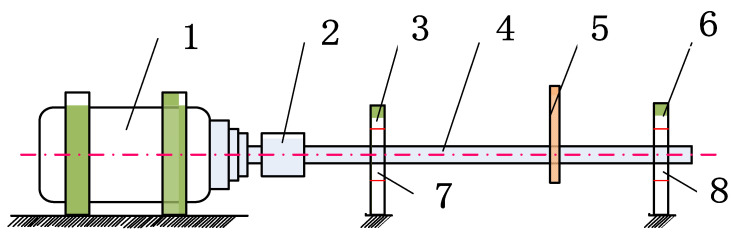
Experimental layout. 1. Motor; 2. coupling; 3. bearing housing I; 4. principal axis; 5. rotors; 6. bearing housing II; 7. bearing I; and 8. bearing II.

**Figure 16 entropy-24-01459-f016:**
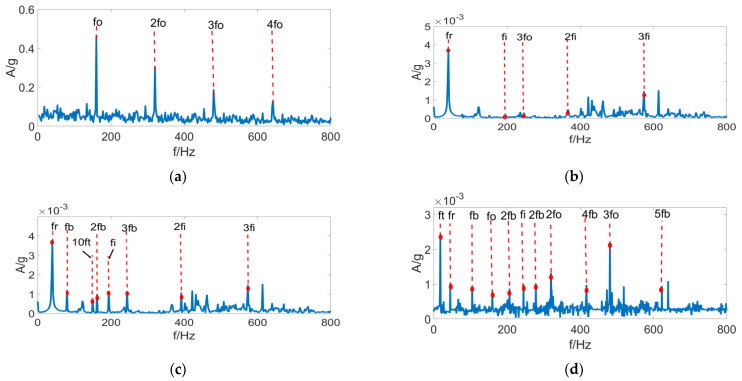
Four methods of envelope results of **(a)** MCKD envelope spectra, (**b**) OMED envelope spectra, (**c**) MOMEDA envelope spectra, and (**d**) FSR-MCReD envelope spectra.

**Table 1 entropy-24-01459-t001:** Simulation-signal parameters.

$f_{0}/\mathrm{Hz}$	$a_{j}$	*M*	*g*	$f_{e}/\mathrm{Hz}$	$\tau_{j}$
50	0.7	1	0.7	10	2%T

**Table 2 entropy-24-01459-t002:** Simulation-signal parameters.

Parameter	Implication	Value	Parameter	Implication	Value
$f_{s}$	Sampling frequency	12,000 Hz	$D_{i}$	Amplitude of shock	Random variable
$f_{i}$	Inner ring	213 Hz	$T_{3}$	Random impact time	Random variable
$f_{o}$	Outer ring	143 Hz	$p_{j}$	Harmonic amplitude	$p_{1}$$=0.03;\;\;\;p_{2}$ = 0.04
$f_{r}$	Rotating frequency	2000 Hz	$f_{4j}$	Harmonic frequency	$f_{41}=45.2$ $;\;f_{42}=56.7$
μ	Ratio coefficient	0.3	$\varphi_{ki}$	Initial phase	Random [−π,π]

**Table 3 entropy-24-01459-t003:** Characteristic frequency of engine bearings.

Rotating Frequency: *fr*	Rolling Element: *fb*	Outer Ring: *fo*	Inner Ring: *fi*	Cage: *ft*
182.9	650.5	1419.5	1843.5	78.8

**Table 4 entropy-24-01459-t004:** Characteristic frequency of bearings.

Rotating Frequency: *fr*	Rolling Element: *fb*	Outer Ring: *fo*	Inner Ring: *fi*	Cage: *ft*
45	89.6	137	222.7	17.01

## Data Availability

Data sharing not applicable.

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
