# Peer review of "Fault Feature-Extraction Method of Aviation Bearing Based on Maximum Correlation Re’nyi Entropy and Phase-Space Reconstruction Technology"

_entropy, 2022, doi:10.3390/e24101459_

Round 1

Reviewer 1 Report

In order to address the difficulty of extracting the features of composite fault signals under low signal-to-noise ratio and complex noise conditions, the authors proposed a feature extraction method based on phase space reconstruction and maximum correlation Re´nyi entropy deconvolution. The proposed method is also verified by simulation, experimental data and bench test. This study is interesting and has a potential to be accepted. Some comments are listed as follows:

(1)   The proposed method combines phase space reconstruction and maximum correlation Re´nyi entropy deconvolution. It is recommended to describe in depth in terms of innovation, whether it is just a simple combination, and make corrections in those places.

(2)   Some academic terms should be unified. For example, composite faults, compound fault, etc.

(3)   Because of too many abbreviations, it is recommended to provide a symbol table.

(4)   The graphics quality and resolution are not high. Please redraw these figures.

(5)   In Fig. 11(b), two “fr” are marked. Please revise this error.

(6)   In Fig. 1, whether the label Sa should be revised as Re? Please check it.

(7)   Some expressions such as CSk1 are not provided. Please check it.

(8)   Please check all formulas to make sure that the symbols in each formula are explained, and try to provide the values used in the paper.

(9)   It is suggested to strengthen the literature research, especially the relevant literature in recent years. At present, the writing of introduction and descriptions still need to be improved. The advantages and disadvantages of the existing methods should be summarized.

(10) Some grammar errors and typos should be carefully revised.

Author Response

Point 1: The proposed method combines phase space reconstruction and maximum correlation Re´nyi entropy deconvolution. It is recommended to describe in depth in terms of innovation, whether it is just a simple combination, and make corrections in those places.

Response 1: The reviewer's comments are significant, and the method is poorly explained, so the author emphasizes it again (and notes it in red) in the first paragraph of the conclusion(Section 5). this method is not a simple combination of PSR and MCReD, but PSR is incorporated into the calculation of MCReD to take advantage of the de-noising and decoupling characteristics of PSR.

Point 2: Some academic terms should be unified. For example, composite faults, compound fault, etc

Response 2:  This article does have such problems and has been revised and marked in red

Point 3: Because of too many abbreviations, it is recommended to provide a symbol table.

Response 3: The symbol table has been added as shown in Table 1

Point 4: The graphics quality and resolution are not high. Please redraw these figures.

Response 4:  All diagrams have changed

 Point 5: In Fig. 11(b), two “fr” are marked. Please revise this error.

Response 5:  Figure 11 has been modified

 Point 6:  In Fig. 1, whether the label Sa should be revised as Re? Please check it

Response6:   Figure 1 is indeed an author oversight and has been revised

 Point 7: Some expressions such as CSk1 are not provided. Please check it

Response 7:  CSk1 detailed derivation has been made in Ref. 18 and references have been cited to illustrate the point.

 Point 8:  Please check all formulas to make sure that the symbols in each formula are explained, and try to provide the values used in the paper

Response 8:  The notation and interpretation of the formula have been checked and revised and marked in red.

 Point 9: It is suggested to strengthen the literature research, especially the relevant literature in recent years. At present, the writing of introduction and descriptions still need to be improved. The advantages and disadvantages of the existing methods should be summarized

Response 9:  Some references are added and explained accordingly( marked in red in section 1).

 Point 10: Some grammar errors and typos should be carefully revised

Response 10:  Syntax and other issues have been checked and modified

Reviewer 2 Report

This paper proposes a fault feature extraction method of aviation bearing based on maximum correlation Renyi entropy, there are still some issues must be resolved.
1. In title,it should be "Renyi entropy"
2.In introduction, some related researches about fault diagnosis based on Renyi entropy should be supplied.
3.Stronger motivation should be considered. Why we must use the Renyi entropy and what's the meaning of \alpha in the proposed method?
4.The quality of Figure must be improved.
5.In conclusion, the boundaries of the proposed method and the future works should be discussed.  

Author Response

Point 1: In title,it should be "Renyi entropy"

Response 1: The problem is indeed an oversight of the author and has been changed

Point 2: .In introduction, some related researches about fault diagnosis based on Renyi entropy should be supplied

Response 2: References to Renyi entropy have been added and marked in red

Point 3: .Stronger motivation should be considered. Why we must use the Renyi entropy and what's the meaning of \alpha in the proposed method

Response 3: The necessity of using Renyi entropy is explained again, in Section 1, the yellow background is added. The meaning of α is explained in Section 2 and marked in red

Point 4:The quality of Figure must be improved.

Response 4: All diagrams have changed

Point 5:.In conclusion, the boundaries of the proposed method and the future works should be discussed.

Response 5: The last paragraph of the paper adds a statement about  the boundaries of the proposed method and the future works

Round 2

Reviewer 2 Report

Authours have addressed my concerns, i recommend to accept the paper.